# Self-Perception of Children and Adolescents’ Refugees with Trauma: A Qualitative Meta-Synthesis of the Literature

**DOI:** 10.3390/bs15121647

**Published:** 2025-11-30

**Authors:** Genta Kulari, Sandra Figueiredo

**Affiliations:** 1Department of Psychology, Universidade Autónoma de Lisboa, 1169-023 Lisboa, Portugal; 2University Research Center in Psychology (CUIP) Integrated in Foundation for Science and Technology, Universidade Autónoma de Lisboa, 1169-023 Lisboa, Portugal

**Keywords:** refugee minors, trauma, mental health, qualitative synthesis, refugee camps, resettlements

## Abstract

Refugee children and adolescents face significant psychological and social challenges, especially in camps or during post-resettlement. We conducted a meta-synthesis of 24 qualitative studies including 870 participants aged 3–19 to explore how they perceive trauma, considering gender, age, and unaccompanied status. Thematic analysis identified five core themes: (1) mental health perceptions, showing reluctance to disclose distress due to stigma and cultural norms; (2) stigma regarding refugee status, reflecting societal prejudice and barriers to integration; (3) desire to belong, including social withdrawal, family cohesion, and religious coping; (4) gender-specific needs, with girls facing early marriage, safety threats, and psychosocial vulnerability; and (5) discrimination from host communities, including verbal, physical, and institutional exclusion. Participants reported pervasive emotional distress, identity conflicts, somatic symptoms, and disrupted social relationships. The findings highlight the complex, multi-layered impact of forced displacement. Thematic analysis proved effective for capturing lived experiences, coping strategies, and contextual influences. These results underscore the urgent need for culturally sensitive, trauma-informed interventions addressing mental health, social support, and protective factors to promote the well-being and integration of refugee minors. The scarcity of research in high-risk camp and detention settings underscores the importance of qualitative inquiry to inform culturally grounded, multi-level psychosocial support.

## 1. Introduction

The global refugee population has steadily increased in size over the past several years, and consequently, receives considerable attention from international media and policymakers. In 2017 the number of refugees worldwide was at an all-time high of 25.4 million ([52]) and by the end of 2024 the number reached 37.9 million refugees, with approximately half of them younger than 18 years old ([52]). Currently, almost a third of all refugees are victims of civil war in Syria (933.000), with large numbers also coming from Afghanistan (558.000) ([52]). At the present, we are witnessing multiple waves of forced migration, particularly from Russia, the Democratic Republic of Congo, Gambia, Israel-Gaza, Syria, Colombia, and Venezuela. Since 2022—and more markedly by 2025—these crises have varied in origin but share the common outcome of mass displacement. Across these cases, whether as refugees or asylum seekers, populations overwhelmingly prefer urban resettlement over camp life, even if resettlement often means staying with extended family ([8]). Yet not all refugees are afforded this opportunity, as host countries may restrict integration and permanent settlement.

European nations, in particular, are now hosting the largest numbers of refugees, attracted by perceptions of greater security and resource availability. This reality highlights an urgent parallel issue: the adequacy of services and resources to assess the mental health of displaced persons, with particular concern for minors, whether accompanied or unaccompanied. Regarding prevalence among refugee children, international estimates revealed markedly elevated rates of mental illness compared with non-refugee peers, especially for Post-Traumatic Stress Disorder (PTSD), Depression and Anxiety Disorders, reaching 87% as maximum for PTSD ([2]; [7]; [55]). One in five refugee children experiences PTSD, and approximately one in seven suffers from depression or anxiety: levels three to five times higher than those seen in general child population ([7]; [40]). Considering different systematic analyses and case reports in previous literature, the overall pattern of high prevalence was consistent across contexts. [2] ([2]) and [7] ([7]) identified multiple interrelated risk factors that increase the vulnerability of refugee and asylum-seeking children to mental illness. These can be grouped into three phases of the refugee experience: pre-migration (conflict and trauma exposure) (1); migration and transit (between countries) (2); and post-migration and resettlement (3).

In the first group, many refugee children have witnessed physical and psychological violence, persecution, displacement, or the death of family members. Therefore, prolonged exposure to war-related trauma leads to neurobiological dysregulation, increasing susceptibility to PTSD and depression; to add, disruption of schooling and normal social development compounds the psychological toll. Respecting the second group, during migration, children often endure unsafe journeys, family separation, detention, or uncertainty about asylum status. These ongoing stressors generate chronic stress responses and feelings of helplessness, which can trigger or exacerbate existing mental health symptoms. The third group refers to the stage after arrival (into the hosting country) when children face acculturation stress, language barriers, discrimination, and economic hardship. These are specific post-migration stressors already well identified in previous investigation, namely limited access to culturally appropriate mental health services (no human resources well prepared in the hosting countries), settlement in camps for long period of time (more than 30 days), prolonged asylum processes (bureaucratic obstacles) and delays in schooling placement ([13]; [18]). In the present study we emphasized that post-migration factors may prolong distress and impede recovery, especially mental health diagnostic and respective follow-up with practitioners, social integration, family stability, and access to education.

Refugee children are at risk considering the above conditions and especially concerning the following circumstances: when they are female, when exposed to cumulative episodes of war and violence, as well when children are displaced with no relatives or friends (unaccompanied) ([31]; [39]; [40]). The type of settlement until now was less focused, and matters considering the likelihood of coping toward PTSD developing among children living in refugee camps for excessive time. Refugee youth perceptions of camp life and resettlement processes remain poorly understood, as mental health assessments are still limited in both contexts. Even when refugees are formally integrated into urban areas, many do not live independently but are instead placed in temporary care institutions under international protection ([4]; [52]; [54]). During prolonged stays in such institutions, or previously in camps, children face heightened risks of trauma, PTSD, and associated maladaptive behaviors, including suicidal ideation ([32]; [42]; [49]).

The vulnerability is particularly critical during childhood and adolescence, when displacement and trauma can profoundly disrupt psychological and developmental processes ([50]). The identified migration stressors specifically related to refugees’ displacement operate through cumulative and mediating mechanisms: they sustain hyperarousal, disrupt emotion regulation, and reinforce feelings of helplessness and alienation ([14]; [24]). Consequently, rather than providing safety and stability, the post-migration environment often perpetuates psychological vulnerability, transforming acute trauma responses into persistent mental health problems such as PTSD, depression, and anxiety. In particular, post-migration stressors act as chronic, compounding mechanisms that prolong or amplify the psychological impact of war-related trauma in refugee children, transforming an acute stress response into persistent psychopathology ([26]; [53]).

This aim of this study is twofold. First, it seeks to explore how refugee children and adolescents (aged 3–19 years) perceive and describe daily stressors within the context of displacement, encompassing both the period spent in refugee camps and the post-resettlement phase, as reported directly by the minors. Second, it examines how these perceptions vary according to gender, age, and unaccompanied status, as these factors have been identified as potential moderators of vulnerability and resilience in previous research. Based on these aims, the following research questions (RQ) were formulated:RQ1: How do refugee children and adolescents perceive and describe their daily stressors during displacement and post-resettlement?RQ2: How are these perceptions influenced by gender, age, and unaccompanied status?

## 2. Materials and Methods

### 2.1. Design

Meta-synthesis is a research method synthesizing findings from existing qualitative studies to build broader interpretations and deeper understanding ([6]; [15]; [30]). One common approach is thematic synthesis, which draws on meta-ethnography and grounded theory ([30]). It involves coding qualitative data conceptually to identify patterns and constructing a model that offers insight into the phenomenon under study ([38]). This method not only enhances theoretical understanding but also informs intervention strategies ([5]). Thematic analysis, a key step in this process, examines data segments obtained from interviews using codes based on recurring semantic features. For example, a code like “trauma” may group together various statements reflecting experiences of being persecuted, even if perceived subjectively by participants. These codes are then analysed to identify key themes and patterns, which can be further examined through tools such as sentiment analysis ([17]; [19]). The process is iterative and comparative, often organizing themes in a hierarchical structure ([30]). To date, no known meta-syntheses have specifically explored how refugee children and adolescents perceive trauma, particularly across different backgrounds, genders, ages, and migration stages—from forced displacement through camp settlement and post-resettlement. Understanding these perspectives is critical to developing age- and context-appropriate support strategies.

### 2.2. Protocol and Search Strategy

Before commencing, we registered our meta-synthesis protocol with [44] ([44]), the international prospective register for systematic review (Appendix A). For this study, we applied an accepted six step method for conducting a meta-synthesis ([30]), consisting of (1) defining the research question and selection criteria, (2) using those criteria to select studies, (3) undertaking a quality assessment, (4) extracting and presenting formal data, (5) conducting data analysis, and (6) reporting the synthesis. The selection of included studies is illustrated in the flow diagram, following the Preferred Reporting Items for Systematic Reviews and Meta-Analyses (PRISMA) framework (Figure 1).

We conducted our search using five electronic databases (PubMed, PsycINFO, CINAHL, Scopus, and Web of Science) from January 2017 to August 2025. These databases were chosen to capture studies from a range of research disciplines, including medicine, psychology, sociology, and anthropology. We searched terms including capturing trauma, post-traumatic stress disorder, mental health, and qualitative research. Also, all keywords’ searches were combined with refugee, displaces person, asylum seeker, and humanitarian entrant. The search terms are identified in Table 1. The characteristics of the selected articles can be seen in Table 2.

### 2.3. Selection: Inclusion & Exclusion Criteria

We screened titles and abstracts of identified studies for eligibility, followed by full text review where indicated, using the following inclusion criteria to identify studies that:(a)Used a qualitative research design such as semi-structured interviews or focus group or mixed-method studies reporting qualitative data;(b)Were published in a peer-review journal, and written in English;(c)Sampled children and adolescents aged 3 to 19 years living in refugee camps or in the advanced phase (resettlements) experiencing aspects of trauma. We chose the age range 3 to 10 to cover definition of childhood and 10 to 19 years to cover WHO definition of adolescents;(d)Qualitative studies that explored perception of children and adolescents’ traumatic experience of settling in a refugee camp and after (resettlement) auto reported by minors;(e)Included participants with self-reported depression or PTSD or diagnosed by a health professional, regardless of severity or treatment received. We included studies that involved participants with depression, PTSD, with or without a comorbid anxiety.

Studies were excluded if they:(a)Sampled participants above the age of 19;(b)Qualitative studies that did not provide perspective on children and adolescents trauma but focused on the experience from other sources such as parents, teachers, healthcare professionals, guardians, etc.;(c)Sampled participants with previous history of depression;(d)Sampled participants with: a co-morbid chronic physical disability; any co-morbid mental illness (e.g., schizophrenia, personality disorders); or a co-morbid neurocognitive disorder (e.g., autism spectrum). This avoids capturing the experience of trauma settling in a refugee camp or resettlement in the context of co-morbid conditions beyond common mental disorders;(e)Studies that included quantitative research design or literature review, conceptual articles, commentaries, books, and book chapters;(f)Studies that focus on studying different contexts other than refugee camp experience (i.e., school context, acculturation, community in a daily basis after resettled);(g)Studies that were not written in English.

Exclusion of grey literature and selection of English publications may be a limitation that we acknowledge in this study.

### 2.4. Data Screening and Extraction

One researcher conducted the search, removed duplicates, and screened the titles and abstracts of all studies for relevance, before assessing the full text of identified studies for eligibility. These results were then independently screened by the second author. The other researcher was assigned for full text assessment to ascertain agreement over inclusion/exclusion, meeting regularly as a group to discuss eligibility.

### 2.5. Quality Appraisal

One researcher appraised all eligible studies for quality using Critical Appraisal Skills Program (CASP), a 10-item quality assessment tool for qualitative research, discussing this with the research team ([43]). Studies were appraised on these items grouped under three categories: validity (clarity of research aims, appropriateness of qualitative methodology, research design, recruitment strategy, and data collection, appropriate consideration of researcher reflexivity), results (ethical consideration, appropriateness of data analysis, clarity of findings stated), and utility (the value of the research). Study characteristics and appraisal criteria were summarized in Table 2. We chose not to exclude studies based on our assessment of low quality, instead, our synthesis of findings took into account our CASP based judgement on the quality of included studies, as suggested by methodological guidance ([30]) and can therefore be interpreted in this context.

### 2.6. Data Analysis

For each included study, one researcher identified any text relating to refugee camp or resettlement experiences in the result section (whether direct quotes or author’s interpretation) and imported relevant passages into a qualitative data analysis software package (NVivo Software model 2020, [45]; electronic dictionaries assured by machine learning—LIWC) to facilitate the process of thematic synthesis. The other researcher independently assessed the studies to identify which passages to import and compare judgments on which data to include or exclude. Having established our final database of extracted qualitative data, one researcher them coded the full dataset, and the other one independently coded data from five randomly allocated studies. Both researchers then compared their coding to develop an initial coding framework. This was then refined through an iterative process, to develop a taxonomy of analytic themes.

## 3. Results

### 3.1. Description of Included Studies

Our search identified 1859 studies, which were reduced to 639 after deleting 1220 duplicates (see Figure 1). Following the titles/abstracts we excluded 251 studies for irrelevance, as they did not meet the study’s inclusion criteria (i.e., systematic review studies, irrelevant population or context such as academic performance or acculturation process). Following the full text review of the remaining 388 studies, a further 366 studies were excluded based on our exclusion criteria leaving a total of 22 studies (see Figure 1). Two studies outside of the original search were added from the list of references in the selected studies, compiling a total of 24 articles (see Figure 1), which we included in this meta-synthesis. We achieved 100% decision agreement on study eligibility between two authors before the discussion. Characteristics of each study are shown in Table 2, including an assessment of study quality using CASP criteria.

The total number of participants was 870, with sample size in each study ranging from 4 to 188. Participants’ ages ranged from 3 to 19 years. Dates of publications ranged from 2017 to 2025, and studies originated from the United Stated (n = 3), United Kingdom (n = 2), Turkey (n = 2), Canada (n = 2), Sweden (n = 2), Germany (n = 2), Portugal (n = 2), Jordan (n = 2), Lebanon (n = 2), South Korea (n = 1), Denmark (n = 1), Netherlands (n = 1), Australia (n = 1), Austria (n = 1).

All studies involved children and adolescents with the status of refugee, either living in refugee camps (n = 3) or resettled in the community (n = 21). Regarding the methodological approach, the analysis of the included studies revealed that thematic analysis is the most widely applied method for investigating the experiences of refugee children and adolescents. Among the 24 studies reviewed, 14 explicitly employed thematic analysis, accounting for approximately 68% of the research. The other studies administered different approaches: grounded theory (3 studies = 14%); semantic and linguistic analysis (2 studies = 9%); content and discourse analysis (2 studies = 9%); and play/art-based coding (2 studies= 9%). Regarding the Critical Appraisal, in specific, the quality observed along the included studies ranges from moderate quality to high quality (5–10). Given the methodological differences across the studies, the 10 item criteria such as “Was there a clear statement of the aims of the research” and “Is the qualitative methodology appropriate?” were answered “yes” and “no”.

### 3.2. Thematic Synthesis

Our thematic synthesis of 24 eligible studies identified 5 analytic themes: (1) perception of mental health, (2) stigma regarding refugee status, (3) the desire to belong, (4) gender needs, (5) discrimination from others. We selected these themes based on the semantic and lexicon frequency of children’s and adolescents’ narratives, considering their perception of refugee status. Syntactics was also valued mainly when the discourse showed high number of repetitions and pauses. Across the discourses, participants showed a marked tendency to evaluate mental health negatively. This formed the basis for the first code, “perception of mental health.”

#### 3.2.1. Theme 1: Perception of Mental Health

The first theme emerged from 13 studies, comprising 341 children and adolescents ranging from 3 to 19 years old. As children and adolescents living in precarious conditions related to their refugee status, they were hesitant about disclosing their feelings of stress, anxiety and depression to people within their social network. They avoided being open about their true selves. Refugee children tended to perceive and describe mental health as a disease or disturbance associated with social isolation and exposure to discrimination. The terminology predominantly used referred to negative illnesses rather than trauma specifically related to their refugee experience. Trauma was rarely explicitly mentioned as a component of mental health across the studies analysed. Despite limited understanding of trauma related to war stressors and forced displacement—key pre- and post-migration stressors—children and adolescents reported depressive symptoms, even without explicitly labelling their feelings as ‘depression.’ This links to subsequent themes, as the impact of forced displacement is associated with difficulties in acculturation, which, in turn, affect the development of a sense of belonging within the host society.

Some individuals described being aware of putting up a façade and of making extensive efforts to maintain this front to avoid talking about their mental health issues.
‘‘*…In some cultures, mental health is not perceived the way that we perceive it in the UK, or America or the Western world. You know, some cultures would just say that you’re a crazy person perhaps, in terms of summing it up. And therefore the stigma associated with that would consequently lead people not to admit it.*”([34])
“*I have so many problems (…) They [psychologists] always want to talk about my past. If I talk about it, it comes again in my brain, it comes again. In the end, maybe I get crazy. I have so many… I do not want to think about it anymore.*”([23])

A range of reasons were given for the non-disclosure of their feelings, summarised in the two sub-themes below.

##### Subtheme 1.1. Mental Health Literacy

Refugee children and adolescents in the 13 included studies commonly expressed that the source of their mental conditions was a result of witnessing violence and loss ([12]; [27]; [33]; [20]; [29]; [35]; [56]; [1]; [51]; [41]; [9]; [34]; [21]; [47]). In these studies, refugee children and adolescents were engaged in a range of different treatments at the time of data collection, which included psychological, activity-based and pharmacological interventions. Participants did not recognize the benefits of talking therapies and their experiences tended to range from passively following instructions to actively finding them unpleasant or unnecessary:
“*…they keep talking to you, that means this person annoying you and you just closing your ears.*”([35])

Additionally, the slow pace of therapy, together with the need to readdress issues from the past were also viewed negatively. Participants shared that these interventions reawakened memories of their country of origin and relocation trauma to their host country that they thought had led to the current mental health concern. This perception has consistently been evidenced as a strong barrier to engagement in the intervention treatment. Hence, interventions were mostly perceived as regressing rather than making progress.
“*…talks doesn’t helps me because every time I was talking about it, it just reminded me about home and was hurting me about same, same problem and more worse.*”.([35])
“*(…) They are asking so many questions, and they do not think that, afterwards, I am going to think about my past. My heart becomes disgusting.*”([23])

Compared to talking therapies, medication and entertainment activities seemed to generate more positive views of treatment in helping them feel better, lifting their mood, stopping their thinking or helping them forget. It appears that for refugee children and adolescents, suppressing their traumatic past experiences and concentrating on recreational activities was the best way of coping. This also helped them avoid being overwhelmed by a combination of distressing past events and an uncertain future. The studies of [35] ([35]) and [3] ([3]) explored that these perceptions were related to participants’ cultures, which elicit shame and guilt to those openly discussing mental health issues.
“*… especially the society where we come from, children don’t talk a lot. Children are not even allowed to reveal things. Children are not even allowed to talk about being abused*.”.([35])

##### Subtheme 1.2. Fear of Being Judged

Four studies ([23]; [33]; [20]; [21]) comprising 69 refugee children and adolescents ranging from 12 to 19 years old reported that minors feared being judged for openly talking about their feelings. Mental health problems were associated with “crazy or mad people”, who were publicly ridiculed, and many times hospitalized in psychiatric services. One study in particular ([34]) explored adolescents’ narratives related to stories of people who suffered mental health problems in their country of origin. Adolescents claimed that mental health problems were associated with lack of hygiene and dirty clothes:
“*‘…Then I told this lady I’m not crazy, I’m not like these, these, you know…I tell her look my hair, look my clothes, I’m not crazy.’*”([34])
“*I mean, I don’t really, uh, express myself, so, yeah. Like, I don’t really tell anyone about how I’m feeling and stuff, you know (…) Uh, I just, uh, I don’t trust anybody.*”([33])

#### 3.2.2. Theme 2: Stigma Regarding Refugee Status

The second theme identified in this study analysis referred to stigma related to refugee status. Stigma consists of negative societal prejudice that intensifies misconceptions of refugees, their integration and well-being. This can lead to difficulties in areas like housing and employment, discouraging help-seeking for mental health issues, and specific concerns related to their future in the hosting country. This second theme, “stigma regarding refugee status” emerged prominently because many children expressed shame about being refugees, which often led to avoidance of discussing their status or circumstances. In some studies, children self-reported anxiety or feelings of unease but tended to conceal these emotions, particularly from authorities or health practitioners. This reflects both a sense of guilt and a perceived lack of social support. Stigma intensified when participants recounted discrimination in public spaces.

This second theme also underscores misconceptions surrounding mental health and trauma in the context of forced displacement. Children and adolescents frequently possessed an incomplete understanding of their circumstances, which were often insufficiently explained to them. They tended to associate refugee status with a negative social identity, thereby reinforcing internalized stigma. Consequently, across 17 manuscripts, four studies comprising 80 children and adolescents ranging from 12 to 19 years old reported concerns related to deportation and difficulties associated with seeking employment ([56]; [51]; [9]; [20]), six studies of 371 minors ranging from 3 to 18 years old reported concerns related to settlement conditions and treatment ([1]; [46]; [11]; [36]; [27]; [22]), and seven studies of 422 minors with ages from 7 to 18 years old reported difficulties with the procedures to attend healthcare services and education institutions ([46]; [11]; [12]; [56]; [10]; [47]; [48]).
“*When I arrived in Portugal, I felt like a criminal… All that confusion… The cops intimidated me a great deal. They said things to me like: “You’re going back to your country.” They said a lot of things like that… They told me to stay at the detention centre with some women and their children. Some women were my mothers….*”([37])

In particular, [9] ([9]) found that unaccompanied refugee children and adolescents lived in highly precarious conditions and received little to no information about the whereabouts or well-being of their family members, a circumstance that substantially intensified their traumatic experiences.
“*They don’t want to say to my custodian that my mother is dead. They were hiding it from me. But later I found out that she is dead. That is it and also here, the time I came here, I used to cry all the time, feeling sad so much and sometimes when I cool down then sometimes it comes again. Sometimes it stops but when someone makes trouble with me, then I used to remind all that things that happened before (…).*”([9])

In contrast to younger children, adolescents reported experiencing greater pressure to find rapid solutions to their employment and housing needs, particularly those approaching the age of 18.
“*I will neither have a quality profession nor a job. That’s it … Whatever we can find….*”([56])
“*I want to know if I will be arrested tonight and have to leave this country or if I can sleep without fear for a few extra months. Please let me retain some control over my own life—13 years old child.*”([51])
“*When we received the papers on [date], we discovered that [sister] was omitted. We knew they were taking her out of our family because she had become an adult. This shocked us.*”([36])
“*The law says it is not written in my papers to work. So this is why I lost this job and now I need a job if I want to stay in this country. I want for me to pay taxes in Austria, think that it can be possible so please I need help. I need, I want to make work and before I work I need to make school and so that I understand the language that is more important. If I can’t speak the language for me to stay in the country it will be difficult. I must speak their language (…),*”([9])

These concerns were also frequently reported by younger children, who often lived in challenging housing conditions and lacked the freedom to engage in everyday activities, such as socializing with peers.
“*We lived [several] years in a camp in [country]… If it’s raining …it would pour on us. When the weather is getting hot the tents burn, because of the electricity … the tents were on fire—Iraqi child.*”([1])
“*I say I want to go out. I want to meet my friends. I want to sleep over with them.**‘No, you have to talk to your assigned legal guardian’, who says no, and the social services say you are a child. So I become sad, and I cry, and I do nothing. I keep it to myself.*”([36])

#### 3.2.3. Theme 3: The Desire to Belong

The third theme reflects refugee youths’ efforts to establish new social connections within the host society, where they frequently reported feeling foreign or out of place. The theme of “desire to belong” was less prominent than the others, likely due to fear of deportation, unsafe living conditions, and economic constraints. Separation from relatives during displacement constituted a major source of distress. Experiences of grief and loss, rather than challenges associated with acculturation, emerged as the primary barriers to engaging with the host community. Perceptions of the host society varied across countries: European contexts such as Austria and Sweden were associated with higher levels of perceived stigma, whereas Australia and several African host countries were described in more positive terms. For many young refugees, survival and safety took precedence over efforts toward social integration.

However, a sense of belonging emerged in a different form, namely increased social cohesion with family members and siblings. Refugee children and adolescents explained that experiencing traumatic events, including the loss of family members and friends as well as prolonged separation, led them to re-evaluate their core beliefs about social relationships. Across eleven studies, 277 participants with ages 8 to 19 years old reported feeling more appreciative of their families and more deeply connected to them when compared to the hosting country community ([29]; [21]; [56]; [51]; [11]) and 111 minors with ages comprised between 8 and 18 years old reported stronger relations with their siblings and relatives ([1]; [10]; [41]; [25]; [47]; [48]). This theme described a number of paradoxes or vicious cycles that were apparent in various forms across these studies classified in two subthemes.
“*Who I am talking to? Rather to my co-citizens, because they are in the same situation as I am and they understand me better. When I am in the company of a German or someone from another country and when I talk a bit, yes, then she or he understands everything, she can listen to me and hear everything I tell, but she cannot feel with me. When I speak with an Afghan person, he has been in the same situation as I am, he can give me better advice about what I can do, how I can create a good life.*”([22])

##### Subtheme 3.1. Fear of Not Belonging

Although refugee children and adolescents described a need or tendency to withdraw socially from peers in the host country, they were also aware that such avoidance could generate or exacerbate feelings of loneliness, shame, humiliation, and fear ([1]; [46]; [22]; [47]).
“*My brother is the first person I go to when I have problems or feel bad. I feel stronger with my brothers and sisters…*”.([56])
“*However, children mostly described missing in- person interactions and were yearning for their relatives.*”.([1])
“*Here there is no equal chance for the blacks and that really sucks and, and racism is not very good here because to me it’s something that always worries me. No friends and especially living in a country which is not an English-speaking country. Many people speak English here but they are always expecting you to speak their language. Even when you can talk to them in their language, someone just looks at you and starts to run away. Someone even looks at you standing far away from you and would just call the police, say there’s a black man here which is all not good.*”.([9])

Some refugee minors described a vicious cycle of loneliness and humiliation arising from their perception of being negatively judged or scrutinized by peers in school or the host society.
“*with my friends in the community, I am more free. But in school, everyone respects one another. But friends in the community, we mess around and laugh. -16 years old).*”.([10])

Fear of stigma frequently motivated social withdrawal. Refugee minors often experienced a tension between the desire to connect with others and the inability to do so. This withdrawal was compounded by limited support from peers, including those within their own community, which further intensified their sense of isolation and alienation.
“*Since I came to Turkey young, my Turkish is very good. However, because I dress differently, it is immediately understood that I am Syrian, and they do not want to make friends with me.*”([56]).
“*I have Syrian friends but feel lonely, and nobody understands me!*”.([56])

In the study of [51] ([51]) participants classified their friends and family support as even more important than professional support:
“*Friends and family are their support system rather than professionals*”.([51])

##### Subtheme 3.2. Religion as a Coping Mechanism

Eleven studies comprising 355 minors with ages ranging from 12 to 19 years old, examined the role of religion as a support system for minors, highlighting its provision of continuity, familiarity (i.e., maintaining connections with their cultural roots), and emotional support ([56]; [10]; [51]; [3]; [9]; [46]; [37]). While not a universal experience, religious practices can become a protective factor for mental health and a valuable coping mechanism, especially in adverse conditions such as refugee status.
“*God is great! He solves all our problems. When I feel bad, when I feel uneasy, praying and praying comforts me.*”([56])
“*Religion was significant source of support, distracted them and prevent distress*”([51])
“*God. Only God. I walked for 5 to 6 days straight, a long time almost without stopping. I no longer felt anything, my feet, my body, nothing. Only God helped. It was because he didn’t want me to die.*”([37])

Some refugee minors also reported that prayer helped them cope with past losses and uncertainty about the future by enabling them to focus on the present.
“*Before bedtime, we were reading the Quran and praying to stay alive for the next day.*”([3])

#### 3.2.4. Theme 4: Gender Needs

An important theme identified in the reviewed articles was gender differences in the perception of refugee status. Four studies comprising 341 minors within age range of 10 to 17 years old, examined the experiences and perspectives of refugee girls ([3]; [46]; [11]; [47]). Gender-specific concerns were evident among refugee youth. Female refugees reported challenges related to future marriage, whereas males were more focused on professional development, career prospects, and financial stability. Coping strategies also differed by gender, with spiritual practices, including prayer, frequently cited as important sources of protection and resilience.
“*A female physician said: ‘Syrian females are suffering more than males from psychosocial problems, especially because of thoughts of being forced to early marriage’.*”([3])

Several studies highlighted key factors shaping girls’ experiences, including safety concerns related to kidnapping, sexual harassment, and child marriage. Despite facing the trauma of forced displacement, fear of deportation, and precarious housing conditions, girls reported experiencing a persistent threat to their physical safety, sometimes even from members of their own family or community.
“*I was pressured at my parent’s place. I didn’t feel any warm-heartedness from my parents and siblings. When a girl is pressured at her parent’s house, she would choose to get married no matter whom the husband is. My parents pressured me a lot, and they watched my every step. They interfered in everything, and they tried to control me in every way.*”([46])

The fear of harassment had several implications for adolescent girls. First, it restricted their mobility, clothing choices, and behaviours, limiting their ability to attend school, visit friends, or access public spaces. Second, parents often sought to marry their daughters at a younger age (typically between 14 and 15 years) as a perceived protective measure against harassment and other risks. Finally, financial precarity led families to arrange early marriages both to reduce the household’s economic burden and to receive a dowry.
“*We don’t have money for food. I want to get married to have a better life. We need money. I need to get married to be able to get what I need.*”([11])
“*I know a girl who’s 20 years old. She’s been married and divorced three times already and her parents keep marrying her by force for the money. She has one child from the first husband, two children form the second husband and she’s already divorced from the third, but she’s pregnant. He divorced her when she was three months pregnant—Older adolescent girl.*”([47])

#### 3.2.5. Theme 5: Discrimination from Others

The final theme identified in this analysis concerned discrimination perceived by refugee minors. This theme encompasses deliberate discriminatory behaviors by members of the host society toward refugees. Discrimination was manifested through verbal bullying and, in some cases, physical aggression in urban public areas across host countries. These experiences of discrimination were reported consistently, regardless of the host country or context. Within refugee camps, concerns were primarily focused on physical safety and basic human rights rather than societal attitudes, resulting in lower engagement with psychological assessments or interventions. By contrast, among resettled urban populations, discrimination was more apparent, taking the form of insults, racism, and barriers related to language proficiency.

In particular, eleven studies comprising 241 minors with ages ranging from 3 to 19 years old, examined in depth the reports of minors regarding discrimination ([12]; [27]; [20]; [21]; [56]; [1]; [10]; [51]; [22]; [25]; [47]). Minor refugees have reported experiencing significant discrimination, including exclusion from education and healthcare, social marginalization, and degrading living conditions. These experiences often result in poor mental and physical health and are frequently rooted in xenophobic and racist attitudes. The study of [12] ([12]) reported:
“*Many participants described damaging attacks on their homes (e.g., explosions, gunshots, other acts of war). In post-resettlement narratives, participants also discussed the vandalism of their homes in their new communities as well as discrimination and harassment at the hands of their neighbours. One participant specifically talked about how someone threw a rock at the participant’s house.*”(p. 497)
“*Once, when I was walking on the street, a 45-year-old man stopped me. He said -You are a Syrian; you don’t pay rent; the state pays you a salary!*”([56])
“*They call us by names like donkey, foal, shoe, tar and so on—Younger adolescent boy.*”([47])

Minor refugees often face social marginalization and a lack of belonging, particularly in educational settings. It is important to note that the studies included in this meta-synthesis report findings from diverse geographical locations, each with different cultural and contextual backgrounds, which may influence the ways refugee needs are addressed.

One study found that, regardless of age, girls tended to speak more positively about their school environment and their relationships with teachers, whereas boys reported more experiences of violence and both verbal and physical harassment ([47]).
“*They didn’t want to see so many Syrian people in [country]. And that’s why we can’t do so many things. For example, this year, when I changed my school, we can’t talk to the [foreign] students. So they think we just have to have a Syrian school. We are separated. And you just think that, we are not normal—Syrian child.*”([1])
“*I don’t go to school here, and it frustrates me. I was a good student. Now, I’m forgetting everything, all the material, I wish I could go back to school and start learning again.*”([46])

In sum, the prevalence of thematic analysis in refugee youth research reflects its methodological robustness, versatility, and ability to capture complex, context-dependent, and multi-layered psychosocial phenomena. Its widespread adoption ensures that qualitative research can accurately represent both the challenges and the adaptive capacities of refugee children and adolescents, making it the preferred approach in this field.

## 4. Discussion

Across the reviewed studies, refugee children and adolescents consistently reported a wide range of psychological, emotional, and social difficulties, reflecting the pervasive impact of forced migration, conflict, and resettlement. Interpreting these findings through both the three-stage refugee experience model (pre-migration, migration, and post-migration) and Bronfenbrenner’s ecological theory allows a deeper understanding of how these stressors are shaped across time and across multiple ecological levels of influence.

Regarding the first research question—How do refugee children and adolescents perceive and describe their daily stressors during displacement and post-resettlement?—the findings demonstrate that reported daily stressors emerged from a cumulative process across all three refugee stages. Pre-migration exposure to violence and loss, migration-phase insecurity, and post-migration instability created an ongoing sense of unpredictability that children and adolescents struggled to fully understand, consistent with the limited mental-health literacy often described in refugee populations. Emotional distress was nearly universal, encompassing anxiety, sadness, fear, anger, and persistent worry about safety and the future ([33]; [56]; [23]). Participants frequently described trauma-related symptoms—intrusive memories, nightmares, hypervigilance, panic, and symptoms consistent with post-traumatic stress—particularly among unaccompanied minors and youths with prolonged exposure to conflict or detention settings ([35]; [27]; [51]). Interpreted ecologically, these symptoms reflect disruptions across multiple systems, including family instability (microsystem), insecure institutional contexts (exosystem), and socio-political stressors (macrosystem).

Considering the second research question—How are these perceptions influenced by gender, age, and unaccompanied status?—this issue was addressed only partially in the literature. While some studies disaggregated data by gender or age (e.g., [3]; [56]), and others focused specifically on unaccompanied minors (e.g., [35]; [9]), the overall evidence remains uneven. Nevertheless, when interpreted within the two theoretical frameworks used in this review, notable patterns emerge. Unaccompanied minors, for example, expressed more intense fears and emotional burden, which aligns with the absence of microsystemic support structures and the heightened vulnerability documented across all three refugee stages. Gender differences, although less systematically explored, intersect with macrosystemic cultural expectations and microsystem-level family dynamics, influencing perceived safety, autonomy, and daily responsibilities.

Social and relational difficulties were frequently reported. Participants described discrimination, bullying, and social exclusion within host communities and peer groups, exacerbating feelings of isolation, loneliness, and marginalization ([46]; [25]; [1]; [28]). From an ecological standpoint, these experiences reflect mesosystem failures—such as weak connections between school, community, and family—and macrosystem-level stigmatization. Many adolescents also experienced role strain and family-related stress, such as premature caregiving responsibilities, separation from parents, or disrupted family structures, contributing to heightened emotional burden and a sense of lost childhood ([36]; [56]). These findings are consistent with developmental theories indicating that displacement amplifies role transitions and undermines normative developmental trajectories.

Several studies highlighted cognitive and identity-related challenges. Youth reported difficulties in understanding and processing their traumatic experiences, confusion regarding migration and resettlement procedures, and persistent worries about personal and family futures ([27]; [12]). Concerns about school integration and language barriers further affected academic goals and confidence, echoing previous research with comparable vulnerable immigrant populations ([16]; [18]). Identity conflicts and perceived stigmatization were recurrent, especially among adolescents navigating cultural integration while maintaining ties to their cultural and religious heritage ([20]; [34]). These challenges align with macrosystemic cultural pressures and the complex identity negotiations typical of adolescence, intensified by forced migration.

Physical manifestations of stress were also evident, with participants reporting headaches, fatigue, and psychosomatic pain associated with anxiety and trauma exposure ([33]; [29]). Educational and institutional environments contributed emotional strain: difficulties with language acquisition, cultural misunderstandings, and academic pressures generated frustration, exhaustion, and reduced self-esteem ([10]; [46]). Within both theoretical models, these academic and somatic difficulties can be understood as downstream consequences of systemic stressors across the displacement trajectory.

In summary, the negative states and symptoms reported by refugee children and adolescents reflect a complex interplay of trauma-related distress, emotional dysregulation, social isolation, identity conflict, and somatic manifestations—processes shaped cumulatively across the pre-migration, migration, and post-migration stages, and across ecological levels ranging from family to sociopolitical systems. These findings underscore the need for multifaceted, trauma-informed, and culturally sensitive interventions that address not only psychological needs but also the relational and environmental determinants of wellbeing.

Regarding methodological considerations, the analysis of included studies reveals that thematic analysis is the most widely applied method for investigating the experiences of refugee children and adolescents. Among the 24 studies reviewed, 14 explicitly used thematic analysis, representing approximately 68% of the research. Other methodological approaches included grounded theory (3 studies = 14%), semantic and linguistic analysis (2 studies = 9%), content and discourse analysis (2 studies = 9%), and play/art-based coding (2 studies = 9%).

The prominence of thematic analysis reflects its adaptability for exploring complex, multifaceted phenomena across diverse populations, contexts, and data types. It was applied to semi-structured interviews, focus groups, narrative inquiries, and creative methods such as drawings, sand play, and group play therapy. For example, studies examining Syrian refugee adolescents in the U.S. and Lebanon applied thematic analysis to capture trauma narratives, coping strategies, and perceptions of mental-health services ([33]; [11]; [46]). Similarly, play-based and art-based interventions with younger children were analyzed thematically to interpret emotional responses, symbolic representations, and experiences of displacement ([29]; [41]).

Several methodological advantages contribute to the widespread use of thematic analysis. It supports the systematic identification and organization of semantic patterns, revealing both explicit accounts of traumatic events and latent psychosocial dynamics such as social support, stigma, cultural adaptation, and resilience. Studies involving unaccompanied minors highlighted themes of agency, discrimination, and coping strategies, with thematic analysis allowing nuanced interpretation of how adolescents perceive therapeutic interventions, social relationships, and community integration ([35]; [9]; [51]).

Furthermore, thematic analysis demonstrates remarkable flexibility across sample sizes and contexts, being effectively used in case studies with as few as four participants and in large-scale studies involving over one hundred adolescents ([11]; [46]). This scalability is essential in refugee research, where age, country of origin, migration trajectories, and resettlement environments are highly heterogeneous. By linking codes and themes directly to participants’ narratives, thematic analysis preserves the authenticity of lived experiences while enabling rigorous interpretation and cross-study comparison.

Finally, thematic analysis facilitates practical application and intervention design, as identified themes can inform psychosocial support strategies, culturally sensitive mental-health services, and resilience-promoting approaches. Across the reviewed literature, researchers consistently emphasized the role of family, peer, and religious networks, alongside activity-based and faith-oriented coping mechanisms, as protective factors identified through thematic coding ([23]; [56]; [3]).

### 4.1. Practical Implications

The evidence synthesized across the included studies carries important implications for practice and policy. First, the predominance of thematic analysis highlights the need for context-sensitive approaches that capture the subjective narratives of refugee children and adolescents. Such methods make visible experiences of trauma, loss, discrimination, and identity negotiation that are often obscured in purely quantitative assessments. Second, the findings consistently point to psychosocial needs that extend beyond clinical interventions, suggesting that service provision should integrate educational support, safe spaces for self-expression, peer networks, and culturally grounded coping mechanisms. Trauma-informed mental health programs should be implemented, including group-based therapy for children and adolescents and individual support for unaccompanied minors, to address anxiety, post-traumatic stress, and emotional dysregulation. Schools and community centers should provide structured language support (in second language and not neglecting home languages of children), tutoring, and culturally adapted educational materials to facilitate academic integration and reduce school-related stress. Social inclusion can be promoted through peer mentorship, extracurricular activities, and culturally sensitive group programs that foster belonging while respecting cultural and religious heritage. Family-centered interventions, including caregiver workshops and psychosocial support, are essential to alleviate role strain, support parenting, and mitigate family-related stress. Health services should integrate psychosomatic assessment and monitor stress-related physical symptoms. Finally, insights derived from thematic analysis should guide the development of interventions that strengthen coping strategies, resilience, and social support networks, ensuring programs are evidence-based, culturally responsive, and tailored to the diverse needs of refugee youth.

The cross-cutting evidence of stigma, unmet psychosocial needs, and disrupted developmental trajectories indicates that mental health services for refugee youth require stronger collaboration between health systems, schools, families, and community organizations. Taken together, these insights advocate for multi-layered, culturally responsive interventions that promote both individual well-being and long-term social inclusion

### 4.2. Limitations

Despite presenting notable themes, our findings should be interpreted carefully due to several limitations. First, we used a meta-synthesis method, meaning that our findings were dependent on the quality of the primary studies. Accordingly, we evaluated the quality of the original studies. Second, due to the nature of meta-synthesis, we did not have first-hand encounters with the participants. To mitigate potential data loss, we carried out thorough and rigorous data extraction process, including an examination of the themes as well as participant remarks. Third, we only included peer-reviewed articles published in specific databases and journals. Additional qualitative investigations relevant to our search yet published in a language other than English or published in avenues (e.g., books) not included in this study may report different findings. Fourth, refugee camps or settlements involve an acculturation process. The primary studies in this meta-synthesis did not report any findings related to participants’ acculturation and how this factor may impact their trauma experience. Finally, one million children were born as refugees only in the last three years ([52]). Hence, the present study did not reflect on the voices of children who were born in refugee camps or settlements in the host country.

## 5. Conclusions

The limited number of studies conducted within refugee camps and detention settings may be partly explained by practical and ethical challenges, such as restricted access, political sensitivities, and difficulties in obtaining informed consent in unstable environments. However, this scarcity is problematic, as these contexts often concentrate the highest levels of trauma and psychosocial vulnerability. Expanding the use of qualitative methodologies, particularly thematic analysis, would be especially valuable introducing approaches that allow for a more nuanced understanding of lived experiences, cultural meanings, and coping strategies that are frequently overlooked in quantitative assessments. Greater investment in qualitative inquiry within camp settings is therefore essential to generate contextually grounded evidence that can inform interventions tailored to the most at-risk populations. Future research should therefore prioritize studies within refugee camps and detention settings, where trauma exposure and psychosocial vulnerability are often most severe, yet remain significantly underrepresented in the current evidence base.

## Figures and Tables

**Figure 1 behavsci-15-01647-f001:**
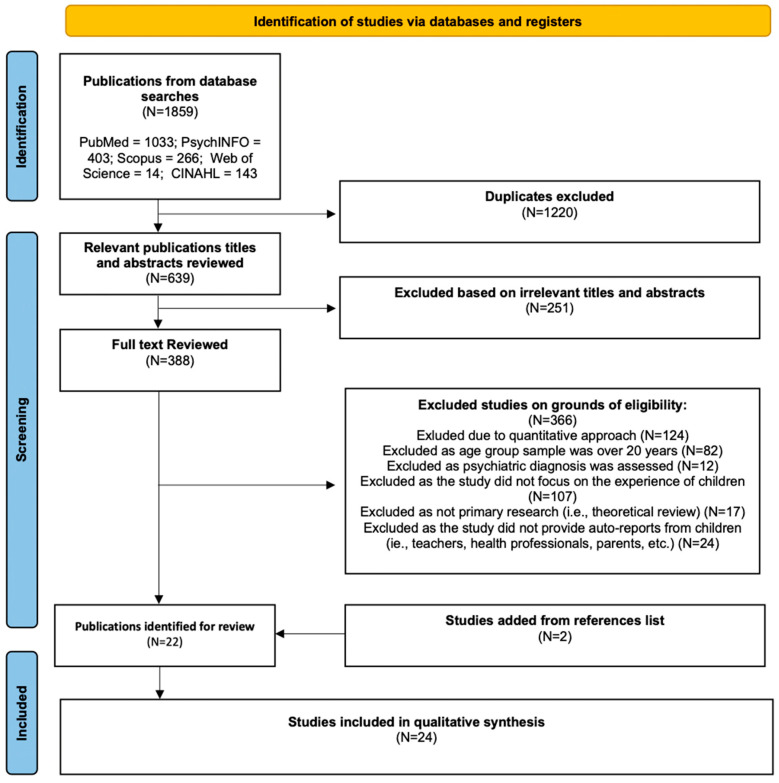
Preferred Reporting Items for Systematic Reviews and Meta-Analyses (PRISMA).

**Table 1 behavsci-15-01647-t001:** List of Search terms.

Search Terms
**Block 1: Target group** Children* OR schoolchild* OR child* OR young* OR adolescent* OR minor*
**Block 2: Trauma experience** Refugee* OR mental health* OR mood* OR psychological health* OR psychological problem* OR externalizing* OR internalizing* OR recreation* OR positive affect* OR life skill* OR well-being* OR family separation* OR friend support* OR stress* OR anxiety* OR distress* OR loneliness* OR stigma*
**Block 3**: **Cognitive and Interpretative responses** Perception* OR think* OR understanding* OR narratives* OR perspective* OR attitude*
**Block 4: Methods** Personal narratives* OR qualitative* OR interview* OR focus group* OR thematic analysis

**Table 2 behavsci-15-01647-t002:** Characteristics of the included studies and Critical Appraisal.

Citation/Country	Sample Size/Population	Aim	Data Collection	Analysis	Themes and Content	Quality Appraisal 1–10
1. [12] ([12]) Country: USA	68 (n = 31 ♀) Aged 7–17, Syrian refugees recruited from primary care during mandatory physical health screening	Disentangle linguistic elements of refugee youths’ trauma narratives and identify biopsychosocial correlates of traumatic stress	Mixed methods analytic approach: Open-ended interview	Semantic analysis of trauma narratives using Linguistic Inquiry and Word Count (LIWC)	1. Witnessing violence and loss 2. Discrimination and bullying 3. Separation and/or social anxiety, panic	7/10 Validity ✓ Clear aims ✓ Appropriate mixed-method approach ✓ Appropriate research design x Appropriate recruitment strategy ✓ Appropriate data collection x Considered reflexivity appropriately Results ✓ Ethical considerations addressed x Rigorous data analysis ✓ Clear statement of findings Utility of results ✓ Value of research
2. [23] ([23]) Country: Denmark	7 (n = 7 ♂) Aged 17–18, unaccompanied Refugee from Middle East and Southeast Asia	Explore unaccompanied refugee adolescents’ perspectives on healing and the mental healthcare offered to them when resettled	Semi—structured interviews and Focus-group interview	Data were coded using cross-sectional indexing	1. Good mental health was associated with social networks and being understood 2. Conventional talk-based therapy re-emerges traumatic events and stigma 3. Healing could come via activities rather than “just talk” (i.e., social, physical, or skill building activities)	8/10 Validity ✓ Clear aims ✓ Appropriate qualitative methodology ✓ Appropriate research design x Appropriate recruitment strategy ✓ Appropriate data collection x Considered reflexivity appropriately Results ✓ Ethical considerations addressed ✓ Rigorous data analysis ✓ Clear statement of findings Utility of results ✓ Value of research
3. [27] ([27]) Country: Canada	35 children (20 families) Ages 3–13, pre-school and school aged children, in a medium-security immigration detention centre	Understand the lived experiences of children held in immigration detention	Narrative inquiry via sand play	Sandtray coding themes	1. Traumatic nature of detention 2. Conflicted understanding of detention and migration 3. Temporal disruption and boredom	7/10 Validity ✓ Clear aims ✓ Appropriate mixed-method approach ✓ Appropriate research design x Appropriate recruitment strategy ✓ Appropriate data collection x Considered reflexivity appropriately Results ✓ Ethical considerations addressed x Rigorous data analysis ✓ Clear statement of findings Utility of results ✓ Value of research
4. [33] ([33]) Country: USA	15 adolescents (n = 10 ♀) Muslim Rohingya refugee adolescents (ages 12–17), living in urban resettlement	Exploring the mental and emotional health challenges of Rohingya adolescents in the U.S.	Qualitative interview	Thematic analysis	1. High emotional distress 2. Self-regulation through internal dialogue and faith-based coping 3. Stigma around mental difficulties and cultural silence 4. Need for safe space for youth expression	6/10 Validity ✓ Clear aims ✓ Appropriate qualitative methodology ✓ Appropriate research design x Appropriate recruitment strategy ✓ Appropriate data collection x Considered reflexivity appropriately Results ✓ Ethical considerations addressed x Rigorous data analysis ✓ Clear statement of findings Utility of results x Value of research
5. [20] ([20]) Country: Canada	7 Adolescents and 8 service providers (gender not provided) Aged 16–19 Syrian refugees	Explore how Syrian refugee adolescents conceptualize mental health through the perspectives of older adolescents and service providers	Grounded theory qualitative exploratory design	Interview comparison	1. Poorly perceived concept of mental health 2. Identity doubt (retain their cultural and linguistic personal background) 3.Personal facts (anxiety and sadness, lack of family and friends’ relation, and stigma from institution impacted negative mental health)	8/10 Validity ✓ Clear aims ✓ Appropriate qualitative methodology ✓ Appropriate research design x Appropriate recruitment strategy ✓ Appropriate data collection x Considered reflexivity appropriately Results ✓ Ethical considerations addressed ✓ Rigorous data analysis ✓ Clear statement of findings Utility of results ✓ Value of research
6. [29] ([29]) Country: South Korea	4 (n = 4 ♀) Aged 8–9 North Korean attending school in South Korea	How the students perceive their traumatic experiences and how they have overcome such difficulties using wisdom and resilience.	case study method	Group play therapy	1. Traumatic life experiences (bringing stressful past events) 2. Need for love and affection (nurturing play) 3. Sense of loss and grief Play therapy improved: 1. Self-empowering to self-expression 2. Regulating powerful emotions	7/10 Validity ✓ Clear aims ✓ Appropriate qualitative methodology ✓ Appropriate research design ✓ Appropriate recruitment strategy ✓ Appropriate data collection x Considered reflexivity appropriately Results ✓ Ethical considerations addressed x Rigorous data analysis x Clear statement of findings Utility of results ✓ Value of research
7. [21] ([21]) Country: Sweden	40 (n = 19♀) Aged 13–19	Exploration of dissociative experiences in multi-traumatized war-refugee youth.	Mixed method	Quantitative analysis and verbal descriptions of mental experiences	1. Common Dissociative experiences 2. Negative self and body perception 3. Depressive mood and emotional dysregulation related 4. Memory disturbances 5. Feelings of non-existence – sense of unreality or loss of presence 6. Suffering around dissociative states	8/10 Validity ✓ Clear aims ✓ Appropriate mixed-method approach ✓ Appropriate research design x Appropriate recruitment strategy ✓ Appropriate data collection x Considered reflexivity appropriately Results ✓ Ethical considerations addressed ✓ Rigorous data analysis ✓ Clear statement of findings Utility of results ✓ Value of research
8. [22] ([22]) Country: Germany	1 (n = 1 ♂) Aged 18 years old, unaccompanied refugee from Afghanistan	Analyse how an unaccompanied minor refugee perceived the psychosocial support he received after arriving in Germany	Case study: Semi-structured interview	Interdisciplinary interpretation group	1. Need for self-determination 2. Perception of hosting country state 3. Link to the hosting country society 4. Personal relationships in the hosting country	8/10 Validity ✓ Clear aims ✓ Appropriate qualitative methodology ✓ Appropriate research design x Appropriate recruitment strategy ✓ Appropriate data collection x Considered reflexivity appropriately Results ✓ Ethical considerations addressed ✓ Rigorous data analysis ✓ Clear statement of findings Utility of results ✓ Value of research
9. [35] ([35]) Country: UK	15 (2 ♀) Aged 15–18 unaccompanied refugees from Afghanistan, Somalia and Iran presenting symptoms of PTSD, depression, and self-harm	Examine mental health services from the perspective of unaccompanied refugee minors and their careers	Semi-structured interviews	Thematic analysis	1. Perception of their practitioners (patient, authority figure, gender and ethnic background were basic barriers) 2. Perception of therapies (Did not recognize the benefits from talking therapies, regressing rather than progress)	8/10 Validity ✓ Clear aims ✓ Appropriate qualitative methodology ✓ Appropriate research design x Appropriate recruitment strategy ✓ Appropriate data collection x Considered reflexivity appropriately Results ✓ Ethical considerations addressed ✓ Rigorous data analysis ✓ Clear statement of findings Utility of results ✓ Value of research
10. [56] ([56]) Country: Turkey	24 (12 ♀) Aged 12–18 years Syrian refugees during the initial settlement	Examine the unique stressors and coping processes of Syrian immigrant youth and the social networks that support them	Focus-group interviews	Ground Theory Framework: Interpretative Phenomenological Analysis	1. Life struggle (unequal working conditions, informal work, cost of living) 2. Peer relationships (discrimination, peer bullying, loneliness) 3. Future anxiety (Fear of deportation, inability to access quality education, disbelief in having a profession, hopelessness for becoming qualified) 4. Social barriers (Prejudices, exclusion, inability to benefit from public institutions) 5. Social support (preferably family and religion)	9/10 Validity ✓ Clear aims ✓ Appropriate qualitative methodology ✓ Appropriate research design ✓ Appropriate recruitment strategy ✓ Appropriate data collection x Considered reflexivity appropriately Results ✓ Ethical considerations addressed ✓ Rigorous data analysis ✓ Clear statement of findings Utility of results ✓ Value of research
11. [1] ([1]) Country: Germany	47 parents 11 children (gender not provided) Aged 8–17 from Syria, Iraq, Palestine, Afghanistan and Eritrea	Explores the perceptions of refugee parents and children experiencing conflict, migration and resettlement to uncover potentially negative and positive influences on the well-being of refugee children	Semi-structured individual and group interviews	Thematic analysis	1. Experiencing disruptions to daily life and structure 2. Exposure to violence that brings harm or deconstruction 3. Facing impediments that obstruct progress 4. dealing with affliction 5. Feeling isolated 6. feeling subjected to rejection	10/10 Validity ✓ Clear aims ✓ Appropriate qualitative methodology ✓ Appropriate research design ✓ Appropriate recruitment strategy ✓ Appropriate data collection ✓ Considered reflexivity appropriately Results ✓ Ethical considerations addressed ✓ Rigorous data analysis ✓ Clear statement of findings Utility of results ✓ Value of research
12. [10] ([10]) Country: USA	7 Adolescents (n = 7 ♂) and 4 parents Aged 14–16 Syrian refugees	Analysis of the acculturation, mental health, and academic experience of Syrian refugee adolescents in the United States	Focus-Group Semi-structured interviews	Thematic analysis	1. English language literacy (difficulty comprehension in classrooms) 2. Ethnic identity (interaction was easy within groups of same values) 3. Interaction teacher-student (their behaviours being considered culturally misunderstood) 4. School culture and expectations 5. Perceived discrimination 6. Mental health issues related to pressure, exhaustion from school and constant tests). 7.Resiliance and hope (support system, and religion)	7/10 Validity ✓ Clear aims ✓ Appropriate qualitative methodology ✓ Appropriate research design ✓ Appropriate recruitment strategy ✓ Appropriate data collection x Considered reflexivity appropriately Results ✓ Ethical considerations addressed x Rigorous data analysis x Clear statement of findings Utility of results ✓ Value of research
13. [51] ([51]) Country: Netherlands	21 (n = 8 ♀) Aged 13–20 diagnosed with PTSD refugees from Middle East, Africa, Eastern Europe, and Asia	Identify factors and processes that according to young refugees promote their resilience	Semi-structured interviews	Grounded theory formulation: Comparing codes	1. Traumatic experience in their country of origin 2. Current stressors such as lack of refugee status 3. Support from their close friends and family and religion 4. Discrimination	10/10 Validity ✓ Clear aims ✓ Appropriate qualitative methodology ✓ Appropriate research design ✓ Appropriate recruitment strategy ✓ Appropriate data collection ✓ Considered reflexivity appropriately Results ✓ Ethical considerations addressed ✓ Rigorous data analysis ✓ Clear statement of findings Utility of results ✓ Value of research
14. [41] ([41]) Country: Turkey	50 (gender not provided) Aged 6–10 refugees from 19 Syrians, and 6 Palestinians, and 25 non-refugee Turkish children	Warfare-and-migration-themed drawings of Syrian and Palestinian children living in Turkey as refugees in comparison to Turkish children’s drawings	Comparative study Art therapy intervention	Thematic analysis	1. Warfare (bombing, soldiers, weapons) 2. Immigration (bus, trains, turtles (living in a suitcase) 3. Death (graves, injured people) 4.flags (identity 5. Hope (heroes) 6. Despair (Crying, sad faces) 7. Nature (sun, clouds, flowers)	5/10 Validity ✓ Clear aims ✓ Appropriate qualitative methodology x Appropriate research design ✓ Appropriate recruitment strategy X Appropriate data collection X Considered reflexivity appropriately Results ✓ Ethical considerations addressed x Rigorous data analysis x Clear statement of findings Utility of results ✓ Value of research
15. [3] ([3]) Country: Jordan	20 primary healthcare professionals 20 Schoolteachers 20 Syrian parents 20 Adolescents (gender not specified) Aged 12–17 Syrian refugees	Assess the psychosocial problems and needs and coping mechanisms of Adolescent Syrian refugees in Jordan	Qualitative study Semi-structured interviews	Thematic analysis	1. Psychosocial problems (stress, depression, isolation, Aggressiveness, Family disintegration, violence, separation from friends) 2. Psychosocial problems gender differences: Syrian adolescents female were at bigger disadvantage then their male counterparts 3. Bullying (verbal and physical forms of bullying) 4. Coping mechanisms 5. Unmet psychosocial and other needs (health services, education, social support)	10/10 Validity ✓ Clear aims ✓ Appropriate qualitative methodology ✓ Appropriate research design ✓ Appropriate recruitment strategy ✓ Appropriate data collection ✓ Considered reflexivity appropriately Results ✓ Ethical considerations addressed ✓ Rigorous data analysis ✓ Clear statement of findings Utility of results ✓ Value of research
16. [46] ([46]) Country: Lebanon	118 (n = 118 ♀) Aged 12–17 Syrian refugees in Lebanon	Married girls may experience additional hardships and thus greater feelings of dissatisfaction in daily life, given their young marriage and responsibilities at home	Mixed methods Auto-recorded Narratives and self-administered questionnaires	Thematic analysis	1.Education (shame, fear, humiliation and loneliness were consistently related to experiences of discrimination when going to school) 2. Safety Concerns (sexual harassment, kidnapping, family forcing) 3. Peer support (family members, friends, Religious support) 4. Longing for life back in Syria	9/10 Validity ✓ Clear aims ✓ Appropriate qualitative methodology ✓ Appropriate research design x Appropriate recruitment strategy ✓ Appropriate data collection ✓ Considered reflexivity appropriately Results ✓ Ethical considerations addressed ✓ Rigorous data analysis ✓ Clear statement of findings Utility of results ✓ Value of research
17. [11] ([11]) Country: Lebanon	188 (n = 188 ♀) Aged 13–17 Syrian adolescents	Explore and describe the self-reported unmet needs of Syrian adolescent girls who migrated to Lebanon between 2011 and 2016	Mixed method Audio-recorded stories	Thematic analysis	1. Unmet needs (housing, sanitation, crowded spaces) 2. Safety needs (stability, protection law, freedom from fear) 3. Need for love and belonging (family, friends, community or within romantic relationships)	9/10 Validity ✓ Clear aims ✓ Appropriate qualitative methodology ✓ Appropriate research design x Appropriate recruitment strategy ✓ Appropriate data collection ✓ Considered reflexivity appropriately Results ✓ Ethical considerations addressed ✓ Rigorous data analysis ✓ Clear statement of findings Utility of results ✓ Value of research
18. [25] ([25]) Country: Australia	19 (n = 13 ♀) Aged 15–18 refugees from 12 different countries	Explore sense of identity and experiences among refugee youth in the context of resettlement	Semi-structured interview and drawing the Tree of Life Method	Thematic analysis	1. Experiencing changes in family roles (loss of parents or relatives) 2. Experience of belonging (discrimination and racial abuse from numerous displacements) 3. Experience of bonds with lost loved ones 4. Dealing with emotions in a new context 5. Experience of self in the context of change	8/10 Validity ✓ Clear aims ✓ Appropriate qualitative methodology ✓ Appropriate research design x Appropriate recruitment strategy ✓ Appropriate data collection x Considered reflexivity appropriately Results ✓ Ethical considerations addressed ✓ Rigorous data analysis ✓ Clear statement of findings Utility of results ✓ Value of research
19. [9] ([9]) Country: Austria	28 (n = 6 ♀) Aged <18 African unaccompanied refugees /Gambia, Somalia, Nigeria, Kenya, Ghana, and Eritrea)	Disclose narrative focuses of traumatised unaccompanied refugee minors in order to further identify the specific needs of this particular subgroup	Structured interview	Thematic analysis	1. War related experiences 2. Present experiences (structure of daily life, personal experience with others, trauma-related levels of distress) 3. Coping strategies (future perspectives, leisure, religion, avoidance & self-control)	8/10 Validity ✓ Clear aims ✓ Appropriate qualitative methodology ✓ Appropriate research design x Appropriate recruitment strategy ✓ Appropriate data collection x Considered reflexivity appropriately Results ✓ Ethical considerations addressed ✓ Rigorous data analysis ✓ Clear statement of findings Utility of results ✓ Value of research
20. [36] ([36]) Country: Sweden	18 (n = 11 ♀) Aged 15–17 refugees from 12 different origins (Bosnia, Syria, Iraq, Iran, United Arab Emirates, Somalia, Eritrea, Gambia, Uganda, Nigeria, Sudan, and Peru)	Capture the shared and varying experiences related to the migration journey and the initial resettlement phase of children recently arriving in Sweden	Semi-structured interviews	Thematic analysis	1. Longing for the good life that cannot be taken for granted a. Experienced of ordinary childhood (hobbies and leisure time, school and education, organizing everyday life) b. Challenging factors (exposure to adversities and violence, family separation, language difficulties, mental health challenges, difficulties in integrating) 2. Challenged agency and changing rights a. The agency is being tested (Intention for immigration) b. Reaching the full age can change everything (rights and regulations for different age groups)	8/10 Validity ✓ Clear aims ✓ Appropriate qualitative methodology ✓ Appropriate research design x Appropriate recruitment strategy ✓ Appropriate data collection x Considered reflexivity appropriately Results ✓ Ethical considerations addressed ✓ Rigorous data analysis ✓ Clear statement of findings Utility of results ✓ Value of research
21. [37] ([37]) Country: Portugal	137 (n = 81 ♀) Aged < 18 years old, from Guinea Conacre (n = 19), Mali (n = 9), Sierra Leone and Congo (n = 8)	Characterize unaccompanied minors and understand the processes of transition into the age of majority using mixed methodologies (quantitative and qualitative), with the help of a survey and autobiographical narratives, as a means of also acknowledging the voice of minors/adults in addressing their trajectories and experiences in the country.	Mixed method: Survey and autobiographical narrative	Thematic analysis	1. Country of origin (reason for leaving, traveling method, 2. Route to the host country (dangers and threats, resourcefulness in the face of risky situations) 3. Reception and initial experiences in the hosting country (detainment at the airport; first impressions of the hosting country) 4. Legal protection measures (relationships with the entity; request for protection) 5. Protective measures in residential care (experience in residencial care, developing meaningful relationships; school integration) 6. Experience of transition to living independently (Precipitation of autonomy; challenges of financial Independence; aspirations for the future)	8/10 Validity ✓ Clear aims ✓ Appropriate qualitative methodology ✓ Appropriate research design x Appropriate recruitment strategy ✓ Appropriate data collection x Considered reflexivity appropriately Results ✓ Ethical considerations addressed ✓ Rigorous data analysis ✓ Clear statement of findings Utility of results ✓ Value of research
22. [34] ([34]) Country: UK	15 (n = 1 ♀) Aged 15–18 unaccompanied refugees from Arab and East African countries	Explore unaccompanied refugee children’s experiences, perceptions and beliefs of mental illness, focusing on stigma.	Semi-structured interviews	Thematic analysis	1. Negative perceptions of the concept of mental health (lose sense of basic upkeep, hygiene, dressing and hair, locked in hospital or prison, sleeps in the street). 2. Anticipated social implication of suffering from mental illness 3. Denial of mental health (alternative explanation avoiding to see a psychologist)	5/10 Validity ✓ Clear aims ✓ Appropriate qualitative methodology x Appropriate research design ✓ Appropriate recruitment strategy X Appropriate data collection X Considered reflexivity appropriately Results ✓ Ethical considerations addressed x Rigorous data analysis x Clear statement of findings Utility of results ✓ Value of research
23. [47] ([47]) Country: Jordan	15 adolescents (n = 11 disabled adolescents and n = 4 married girls) Aged 10–17 years Syrian refugees in Azraq camp	Exploring the experiences of younger (10–12 years) and older (15–17 years) adolescent girls and boys in Azraq camp in four capability domains: education, voice of agency, bodily integrity and freedom of violence, and psychological well-being	Mixed-method longitudinal design: quantitative research with 4000 adolescents and caregivers and qualitative research with 220 adolescents, caregivers and key informants	Thematic analysis	1. Education and learning 2. Bodily integrity and freedom from violence (girls report fear of harassment and kidnapping) 3. Voice and agency (conservative social norms given specific to girls regarding movement, acting, dressing code, marriage. 4. Psychosocial wellbeing (painful memories of the war, its impact in them and their siblings) 4. Support system and role model (close relationships with their parents and siblings 5. Access to technologies 6. Mobility in Azraq Camp	7/10 Validity ✓ Clear aims ✓ Appropriate qualitative methodology ✓ Appropriate research design x Appropriate recruitment strategy ✓ Appropriate data collection x Considered reflexivity appropriately Results ✓ Ethical considerations addressed ✓ Rigorous data analysis x Clear statement of findings Utility of results ✓ Value of research
24. [48] ([48]) Country: Portugal	9 (n = 6 ♀) Aged 15–18 years old refugee adolescents from 9 different cultural background (Congo, Guinea–Bissau, Senegal, Algeria, Benin, Morocco, Mali, Gambia, and Afghanistan)	Shed light on: the various barriers the minors struggle with during the process of hosting and inclusion, the obstacles we face on our side while conducting research, and art as a dialogue facilitator between cultures and a therapeutic supplementary tool for inclusion	Qualitative design Workshops: drawing and photography	Content analysis	1. Diversity of background and projects 2. Artistic/creative workshops as a useful tool to express emotions, perceptions and hopes 3. Focus on future & Inclusion rather than just trauma	4/10 Validity x Clear aims ✓ Appropriate qualitative methodology x Appropriate research design ✓ Appropriate recruitment strategy x Appropriate data collection x Considered reflexivity appropriately Results ✓ Ethical considerations addressed x Rigorous data analysis x Clear statement of findings Utility of results ✓ Value of research

## Data Availability

No new data were created or analyzed in this study.

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
