# Peer review of "Self-Perception of Children and Adolescents’ Refugees with Trauma: A Qualitative Meta-Synthesis of the Literature"

_behavsci, 2025, doi:10.3390/bs15121647_

Round 1

Reviewer 1 Report

Comments and Suggestions for Authors

This is an interesting study, exploring an important and relevant topic: the wellbeing of forcibly displaced children. However, I have several concerns, namely: 1) theoretical rationale (which is underdeveloped); 2) the clarity of the research questions; and 3) the used methodology (screening process and the aggregation of results across contexts). Additionally, the manuscript should be carefully checked on language problems (preferably by a native speaker).

Introduction

-In the introduction, there is only one brief paragraph about the mental health of children in refugee camps. I would expert a much more developed theoretical framework here: for example, why are children affected (prevalence estimates?)/ what makes that they are so at-risk, and what are explanatory mechanisms? Which circumstances make that the risks for their healthy development are unacceptable high? Also: what do we know about forced migration and mental health, and post-migration stressors?

- Relatedly, I think the RQ should be further developed and motivated (final paragraph).

Materials and Methods

  • Please check the text on problems with language. For example, ‘synthesizes’ should be ‘synthesizing’
  • Why was the abstract screening not performed by a second reviewer, as common guidelines prescribe?
  • Quality appraisal: please provide a reference to the CASP in the first sentence.
  • Please also add a reference to the NVivo software.

Results

  • Description of included studies, please correct language problems here. Also: please explain main reasons for excluding studies based on title and abstract, and also based on full text. Furthermore, what reference list are you referring to?
  • Please explain whether the 100% agreement was reached before or after discussion of specific articles.
  • When describing the # of participants, please correct language problems in this section (also a full stop is missing), such as sample sizes and The Netherlands.
  • What do you mean with the sentence, ‘In what respects…’ Please rephrase.
  • Please give more details about the results of the critical appraisal.
  • In general in the Results section, please correct language problems, such as ‘copping’.
  • I think the citations are very powerful, but extracted from the same sources. Could you vary the sources more, also within the themes that you describe (i.e. list citations from different articles)?
  • The title of the second them ‘Stigma regarding refugee awareness’ is a bit puzzling to me. Why not using ‘Stigma regarding refugee status’ for example?
  • Table 2 is not in line with APA guidelines and should be formatted in landscape orientation.
  • Before describing the themes, a thorough description of the context/setting of the included articles would be helpful (country, number of participants per study, age of the children, etc.) This would help to be able to interpret the resulting themes.
  • The study is focused on refugee camps, but refugee camps in Europe are very different from the setting in non-Western countries. Please account for these differences in your theoretical framework (introduction) and in the Results section.

Discussion

  • Please start the Discussion with the aim of the research project.
  • The Discussion reads more as sum-up of what has been found in in the included studies, instead of an integration of the results in a previously established theoretical framework. Could the authors integrate their findings in the existing body of literature?
Comments on the Quality of English Language

The manuscript should be checked carefully on language problems.

Author Response

Reviewer comments

Thank you for giving us the opportunity to resubmit a revised draft of the manuscript “Self-perception of children and adolescents’ refugees with trauma: a qualitative meta-synthesis of the literature” for publication in the Journal of Behavioral Sciences. We appreciate the time and effort the reviewers have dedicated to providing feedback on the manuscript and we are grateful for the insightful comments and valuable improvements to the paper. We have incorporated all of the suggestions made by the reviewers.

Reviewer 1

Point 1: Comment
This is an interesting study, exploring an important and relevant topic: the wellbeing of forcibly displaced children. However, I have several concerns, namely: 1) theoretical rationale (which is underdeveloped); 2) the clarity of the research questions; and 3) the used methodology (screening process and the aggregation of results across contexts). Additionally, the manuscript should be carefully checked on language problems (preferably by a native speaker).

Response- Thank you for your comment and very insightful suggestions. We have incorporated all of the suggestions made by the reviewer in the revised manuscript.

Point 2. Introduction

Comment 1.

-In the introduction, there is only one brief paragraph about the mental health of children in refugee camps. I would expert a much more developed theoretical framework here: for example, why are children affected (prevalence estimates?)/ what makes that they are so at-risk, and what are explanatory mechanisms? Which circumstances make that the risks for their healthy development are unacceptable high? Also: what do we know about forced migration and mental health, and post-migration stressors?

Response 1. Many thanks for pointing this out, information was carefully added along pages 3 and 4 to provide substantial response to each question made in this point – prevalence numbers and more explicit factors associated to the theoretical framework underlying stressors (groups), risk (effects), circumstances, correlation between variables, and explanatory mechanisms, and taking into account more evidence to justify the above.

Comment 2. Relatedly, I think the RQ should be further developed and motivated (final paragraph).

Response 2. We appreciate this comment, and the detail was incorporated at the bottom of Introduction, adding also 2 RQ as they also appear in Discussion section. We hope these changes can provide more clarity.

Point 3. Materials and methods

Comment 3. Please check the text on problems with language. For example, ‘synthesizes’ should be ‘synthesizing’

Response 3. Thank you for your comment. As identified by the reviewer we have corrected the indicated errors, and the revised manuscript was proof-read.

Comment 4. Why was the abstract screening not performed by a second reviewer, as common guidelines prescribe?

Response 4. Thank you for pointing this out. We apologize for the unclear description of Data Screening section. The first author did the initial abstract screening, and the second author independently screened these results for eligibility. We have added this information in the revised manuscript.

Comment 5. Quality appraisal: please provide a reference to the CASP in the first sentence.

Response 5. Thank you for your comment. The revised manuscript has added the reference manuscript of CASP

Comment 6. Please also add a reference to the NVivo software.

Response 6. The revised manuscript provided the reference as suggested by the reviewer.

Point 4. Results

Comment 7. Description of included studies, please correct language problems here. Also: please explain main reasons for excluding studies based on title and abstract, and also based on full text. Furthermore, what reference list are you referring to?

Response 7. Thank you for your comment. As rightfully indicated by the reviewer, the revised manuscript provides a reason for exclusion studies based on titles and abstract. The description regarding the exclusion of the full text it is identified in the Prisma Flow figure. The reference list refers to the bibliography in the end of the selected articles. The revised manuscript has reformulated the sentence to provide clearer meaning behind the section criteria.

Comment 8. Please explain whether the 100% agreement was reached before or after discussion of specific articles.

Response 8. We are grateful to the reviewer’s comment The revised manuscript has reformulated the sentence to provide a clearer concept. The agreement was reached before the discussion of each article.

Comment 9. When describing the # of participants, please correct language problems in this section (also a full stop is missing), such as sample sizes and The Netherlands.

Response 9. Thank you for your suggestion. The revised manuscript has reformulated the manuscript to comply with the comments and correct the errors.

Comment 10. What do you mean with the sentence, ‘In what respects…’ Please rephrase.

Response 10. Thank you for pointing this out. The revised manuscript has undergone a further proofreading check in order to increase the quality of the language.

Comment 11. Please give more details about the results of the critical appraisal.

Response 11. As requested by the reviewer the revised manuscript provides further details on the results from critical appraisal.

Comment 12. In general in the Results section, please correct language problems, such as ‘copping’.

Response 12. We are very grateful to the reviewer comments. The revised manuscript has undergone a very thorough proof-reading

Comment 13. I think the citations are very powerful, but extracted from the same sources. Could you vary the sources more, also within the themes that you describe (i.e. list citations from different articles)?

Response 13. Thank you for your comment. Indeed, the articles are repetitive due to their very similar findings. However, the result section elaborates themes from the 24 selected studies, and therefore, we believe that including different articles outside of the inclusion criteria would contradict the context of the results.

Comment 14. The title of the second them ‘Stigma regarding refugee awareness’ is a bit puzzling to me. Why not using ‘Stigma regarding refugee status’ for example?

Response 14. We appreciate the suggestion of the reviewer and as indicated the revised manuscript had reformulated the second theme.

Comment 15. Table 2 is not in line with APA guidelines and should be formatted in landscape orientation.

Response 15. Thank you for the comment. During the first submission, we included the table in a landscape orientation, however, the journal organized the table in vertical orientation.

Comment 16. Before describing the themes, a thorough description of the context/setting of the included articles would be helpful (country, number of participants per study, age of the children, etc.) This would help to be able to interpret the resulting themes.

Response 16. Thank you for your comment. We appreciate the very insightful suggestion of the reviewer and have provided further information regarding the age and number of participants for each theme.

Comment 17. The study is focused on refugee camps, but refugee camps in Europe are very different from the setting in non-Western countries. Please account for these differences in your theoretical framework (introduction) and in the Results section.

Repones 17. We appreciate the reviewer comment. We agree with the reviewer comment and have provided more information regarding the refugee comps conditions based on their location in the theoretical background and in the results section. All the changes can be tracked with yellow highlight.

Point 5. Discussion

Comment 18. Please start the Discussion with the aim of the research project.

Response 18. Many thanks. The two research questions were inserted at the beginning of Discussion along with the results’ perspective or conclusion. One aim is divided across two clear RQ.

Comment 19. The Discussion reads more as sum-up of what has been found in in the included studies, instead of an integration of the results in a previously established theoretical framework. Could the authors integrate their findings in the existing body of literature?

Response 19. Thank you for your insightful comment. As suggested by the reviewer, the revised manuscript had reformulated the discussion.

Reviewer 2 Report

Comments and Suggestions for Authors

This is a very important topic and a very important study. I found it to be particularly compelling because the authors undertook a meta-analysis of qualitative studies of children and adolescent refugees who had lived in refugee camps, providing us with an overview of the state of the research on this sensitive topic - which reflects a world-wide crisis . Thus, we learn that there are only 24 such studies, one indication that this topic is clearly under studied. this article, then, provides us with important information and understandings, providing guidelines for what else needs to be researched about such children/youth and also supplying ideas for psycho-social treatment for them that is relevant to their needs. So, I am very glad that such a study was undertaken.

Nevertheless, the manuscript has a few problems that need to be fixed before publication.

  1. The subtheme 1.1. - lack of literacy - unclear why the authors gave this subtheme this name. Please explain/clarify or change so that it matches what comes under this heading.
  2. I would also rephrase subtheme 3.2 - tie to religion.  this section focuses on religious belief/observance as a coping mechanism. the title should reflect this
  3. it is very difficult to read Table 2. Please format it in a way that it is much more comprehensible.
  4. There is uneeded repetition in the article in a few places. I recommend cutting down/deleting the repetitions. This ties into the Practical Implications section - I would use the number of words that were deleted in the repetitions to instead be used to provide more concrete ideas for the Practical section.  

these are my major comments. 

Comments on the Quality of English Language

The article needs to be edited - especially after the Methods section. There are a number of typos, missing punctuation and words. 

Author Response

Reviewer comments

Thank you for giving us the opportunity to resubmit a revised draft of the manuscript “Self-perception of children and adolescents’ refugees with trauma: a qualitative meta-synthesis of the literature” for publication in the Journal of Behavioral Sciences. We appreciate the time and effort the reviewers have dedicated to providing feedback on the manuscript and we are grateful for the insightful comments and valuable improvements to the paper. We have incorporated all of the suggestions made by the reviewers.

Reviewer 2

Point 1: Comment
This is a very important topic and a very important study. I found it to be particularly compelling because the authors undertook a meta-analysis of qualitative studies of children and adolescent refugees who had lived in refugee camps, providing us with an overview of the state of the research on this sensitive topic - which reflects a world-wide crisis . Thus, we learn that there are only 24 such studies, one indication that this topic is clearly under studied. this article, then, provides us with important information and understandings, providing guidelines for what else needs to be researched about such children/youth and also supplying ideas for psycho-social treatment for them that is relevant to their needs. So, I am very glad that such a study was undertaken. - Nevertheless, the manuscript has a few problems that need to be fixed before publication.

Response - Thank you for your very positive comment regarding our initiative to publish our work. We have incorporated all the suggestions indicated by the reviewer in the revised manuscript

Comment 1. The subtheme 1.1. - lack of literacy - unclear why the authors gave this subtheme this name. Please explain/clarify or change so that it matches what comes under this heading.

Response 2. We are grateful to the reviewer comment. We agree with the reviewer and have reformulated the subtheme

Comment 2. I would also rephrase subtheme 3.2 - tie to religion.  this section focuses on religious belief/observance as a coping mechanism. the title should reflect this

Response 2. We agree with the very insightful suggestion of the reviewer and have reformulated the subtheme

Comment 3. it is very difficult to read Table 2. Please format it in a way that it is much more comprehensible.

Response 3. Initially to our submission we provided the table in a landscape format, however, the journal format restructured the table within this format.

Comment 4. There is uneeded repetition in the article in a few places. I recommend cutting down/deleting the repetitions. This ties into the Practical Implications section - I would use the number of words that were deleted in the repetitions to instead be used to provide more concrete ideas for the Practical section.  

Response 4. Thank you for pointing this out. Revised accordingly and the practical implications were now specified for detailed contexts of refugee children in different settlement cases. One small part of this section was removed because sounds like ‘Limitations’ of the study and these are already identified (to avoid repetition).